# Multi-objective simulated annealing for hyper-parameter optimization in convolutional neural networks

Ayla Gülcü and Zeki Kuş

Computer Science, Fatih Sultan Mehmet University, Istanbul, Turkey

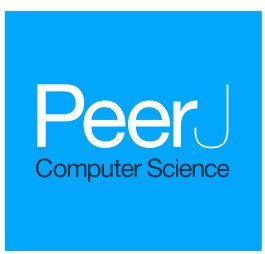

## ABSTRACT

In this study, we model a CNN hyper-parameter optimization problem as a bi-criteria optimization problem, where the first objective being the classification accuracy and the second objective being the computational complexity which is measured in terms of the number of floating point operations. For this bi-criteria optimization problem, we develop a Multi-Objective Simulated Annealing (MOSA) algorithm for obtaining high-quality solutions in terms of both objectives. CIFAR-10 is selected as the benchmark dataset, and the MOSA trade-off fronts obtained for this dataset are compared to the fronts generated by a single-objective Simulated Annealing (SA) algorithm with respect to several front evaluation metrics such as generational distance, spacing and spread. The comparison results suggest that the MOSA algorithm is able to search the objective space more effectively than the SA method. For each of these methods, some front solutions are selected for longer training in order to see their actual performance on the original test set. Again, the results state that the MOSA performs better than the SA under multi-objective setting. The performance of the MOSA configurations are also compared to other search generated and human designed state-of-the-art architectures. It is shown that the network configurations generated by the MOSA are not dominated by those architectures, and the proposed method can be of great use when the computational complexity is as important as the test accuracy.

## INTRODUCTION

Convolutional Neural Networks (CNNs) differ from multi-layer perceptron models with the use of convolution operators instead of matrix multiplications in at least one of its layers (*LeCun et al., 1990*; *LeCun et al., 1998*; *Goodfellow, Bengio & Courville, 2016*). Excellent results obtained for object classification problems in ILSVRC (IMAGENET Large Scale Vision Recognition Competition) accelerated the use of these networks in other vision related problems like face and activity recognition (*Russakovsky et al., 2015*). With the availability of increasing computational resources, winning architectures of the competition, aka state-of-the-art models, became deeper and deeper resulting in very high classification accuracy rates. In 2014, ILSVRC winner model, VGGNet (*Simonyan & Zisserman, 2014*), had 19 layers, but the following years' state-of-the-art models, for example, ResNet (*He et al., 2016*) and DenseNet (*Huang et al., 2017*) had over 100 layers.

Corresponding author
Ayla Gülcü, agulcu@fsm.edu.tr

Finding a CNN architecture that generates high quality results for a given problem is a challenging process which requires a systematic approach rather than by trial and error. Moreover, trying all possible parameter combinations is infeasible due to the size of the parameter space. The problem of automatically designing CNN architectures and selecting the best set of hyper-parameters for the network has drawn attention of many researchers over many years. It is shown in many studies that using algorithmic approaches rather than manual tuning process results in simpler networks with improved classification performance (*Bergstra & Bengio, 2012*; *Real et al., 2017*; *Ma et al., 2020*). This automated neural architecture search (NAS) and hyper-parameter optimization is not only limited to CNNs, but also applicable for both feed-forward and Recurrent Neural Networks (RNNs). For example, the architectures and the hyper-parameters of Long Short-Term Memory networks which are the most widely-used variants of RNNs can be optimized with the proposed approaches.

Hyper-parameter optimization (HPO) is the most basic task in automated machine learning (AutoML) which reduces the human effort by automatizing the labor intensive hyper-parameter tuning process. With HPO, a wider solution space can be searched, which in turn may yield in better performing configurations for the problem at hand (for a detailed view on HPO, please refer to *Feurer & Hutter, 2019*). On the other hand, NAS is a specialized hyper-parameter optimization problem which involves discrete hyper-parameters as in HPO, but with an additional structure that can be captured with a directed acyclic graph (DAG) (*Li & Talwalkar, 2020*). In NAS methods, the search space for designing an entire architecture contains too many nodes and edges; therefore it is usually defined over smaller building blocks called *cells* which drastically reduce the search space. A new architecture is then built by stacking these cells in a predefined manner. The number of the cells and the types of those cells along with the types of connections allowed among those cells are among important design decisions in NAS studies (*Elsken, Metzen & Hutter, 2019*).

Random Search (RS) (*Bergstra & Bengio, 2012*), Bayesian Optimization (BO) approaches (*Hoffman & Shahriari, 2014*) and population-based optimization algorithms such as Genetic Algorithms (GAs) (*Holland, 1992*), Evolutionary Algorithms (EAs) (*Back, 1996*) and Particle Swarm Optimization (PSO) (*Eberhart & Kennedy, 1995*) have been successfully used to select the best CNN hyper-parameters. Especially, EAs are among the most widely used techniques for HPO. In (*Real et al., 2017*; *Suganuma, Shirakawa & Nagao, 2017*; *Elsken, Metzen & Hutter, 2018*) EAs have been used for optimizing the network architecture and gradient-based methods have been used for optimizing the network weights. Single-stage algorithms like Simulated Annealing (SA) and its variants have also proven to be effective for CNN HPO problems (*Gülcü & Kuş, 2020*).

NAS has become a very important research topic especially after it has obtained competitive performance on the CIFAR-10 and Penn Treebank benchmarks with a search strategy based on reinforcement learning (*Zoph & Le, 2016*). However, the computational cost of this approach led the researchers to seek other methods with less computational requirement but with higher classification performance. Relaxation-based methods (*Liu, Simonyan & Yang, 2018*) try to improve the computational efficiency of

NAS approaches, but these approaches still require huge computational resources. EAs provide a good alternative for NAS, but they still suffer from the large computational requirement (*Liu et al., 2018*). On the other hand, it is shown that EA-based NAS methods that use limited search budgets result in poor classification performance (*Xie & Yuille, 2017*; *Sun et al., 2019a*). Therefore, recent HPO and NAS approaches not only consider the error rate, but also the complexity brought by the proposed configuration or architecture.

A single-objective optimization problem involves a single objective function, and usually results in a single solution. However, in many real-life problems, there are multiple objectives to be considered, and these objectives usually conflict with each other. Optimizing a solution with respect to one objective often results in unacceptable results with respect to the other objectives. For example, achieving a network with a low error rate usually comes with a huge cost of computational complexity. Thus, a perfect multi-objective solution that simultaneously optimizes each objective function is almost impossible. A minimization multi-objective optimization problem with $K$ objectives is defined as follows (*Konak, Coit & Smith, 2006*): Given an $n$-dimensional decision variable vector $x = \{x_1,..,x_n\}$ in the solution space $X$, find a vector $x*$ that minimizes a given set of $K$ objective functions $z(x*) = \{z_1(x*),..,z_K(x*)\}$. The solution space $X$ is generally restricted by a series of constraints and bounds on the decision variables. There are two general solution approaches to Multi-Objective Optimization (MOO). One is to combine all of the objective functions into a single composite function using methods like weighted sum method, but in practice it is very difficult to select the proper weights that will reflect the decision maker's preferences. The second approach is to provide the decision maker a set of solutions that are non-dominated with respect to each other from which she/he can choose one. A reasonable solution to a multi-objective problem is to find a set of solutions, each of which satisfies the objectives at an acceptable level without being dominated by any other solution. A feasible solution $x$ is said to dominate another feasible solution $y$, $x \succ y$, if and only if, $z_i(x)z_i(y)$ for $i = 1,..,K$ and $z_j(x) < z_j(y)$ for at least one objective function $j$. A solution is said to be *Pareto optimal* if it is not dominated by any other solution in the solution space. Improving a Pareto optimal solution with respect to one objective is impossible without worsening at least one of the other objectives. The set of all feasible non-dominated solutions in the solution space is referred to as the *Pareto-optimal set*, and for a given Pareto-optimal set, the corresponding objective function values in the objective space are called the Pareto-front. In multi-objective optimization, there are two tasks to be completed, where the first task being the optimization task for finding the Pareto-optimal set, and the second task being a decision making task for choosing a single most preferred solution from that set which involves a human interaction. Generating the Pareto-optimal set is often infeasible, and the methods like evolutionary algorithms (EAs) usually do not guarantee to identify the optimal front, but try to provide a good approximation; that is, a set of solutions whose objective vectors are not too far away from the optimal objective vectors. Since the Pareto-optimal set is unknown, comparing the approximations generated by

different methods is a difficult task requiring appropriate quality measures to be used. There are several measures in the literature each of which reflects a different aspect of the quality such as the closeness to the optimal front and the diversity of the solutions in the front (please refer to *Zitzler et al., 2003* for a detailed review on MOO performance assessment).

Multi-objective EA-based approaches are among the most widely-used methods for CNN MOO problems. In these studies, high classification performance is considered as the first objective, and the low computational requirement is considered as the second objective. It is aimed to generate networks that are satisfactory with respect to both of these objectives. (*Kim et al., 2017*) try to optimize the deep neural networks in terms of two competing objectives, speed and accuracy using GAs. In their study, LeNet (*LeCun et al., 1998*) is chosen as the initial architecture, and the performance of the initial solution is improved considering two objectives. Test results based on the MNIST (*LeCun et al., 1998*), CIFAR-10 (*Krizhevsky & Hinton, 2009*) and Drowsiness Recognition (*Weng, Lai & Lai, 2016*) show that the proposed approach yields in models with better accuracy and speed than the initial model. In *Elsken, Metzen & Hutter (2018)*, Lamarckian Evolution and multi-objective EA is used to generate computationally efficient CNNs with high classification accuracy values. Based on the results on CIFAR-10 dataset, the proposed approach achieves competitive results with other multi-objective approaches. *Lu et al. (2019a)* present another multi-objective EA which they call NSGANet that try to optimize both error rate and the number of Floating Point Operations (FLOPs). According to the test results obtained using CIFAR-10, CIFAR-100 and human chest X-rays data sets, the proposed approach increase the search efficiency. Same objectives are also adopted in the study of *Wang et al. (2019)* in which PSO is used to fine tune the hyper-parameters. In the study, best models are compared to DenseNet-121 in terms of both objectives. The results based on CIFAR-10 dataset state that the models generated by the proposed approach dominate the base model.

In this study, we present a single-stage HPO method to optimize the hyper-parameters of CNNs for object recognition problems considering two competing objectives, classification accuracy and the computational complexity which is best measured in terms of FLOPs. For this bi-criteria optimization problem, we use a Multi-Objective Simulated Annealing (MOSA) algorithm with the aim of generating high quality fronts. CIFAR-10 dataset is selected as the benchmark, and the final fronts generated by the proposed algorithm is compared to the fronts generated by a single-objective variant of the same algorithm with respect to several front metrics such as generational distance, spacing and spread. According to the results obtained using these front evaluation metrics, it can be concluded that the MOSA algorithm is able to search the objective space more effectively than the SA method. After performing longer training on the selected configurations, the results again reveal that the MOSA performs better than the SA under multi-objective setting. The MOSA configurations are also compared to human engineered and search generated configurations. When both test accuracy and the complexity are taken into account, it is shown that the network configurations generated by the MOSA are

not dominated by those configurations, and the proposed method can be of great use when the computational complexity is as important as the test accuracy.

# MATERIALS AND METHODS

In "CNNs—An Overview", we remind key concepts of CNNs and multi-objective aspects of CNNs. In "SA Algorithm", we first describe a single-objective SA algorithm, and then in "MOSA Algorithm", we give some details about the MOSA algorithm that discriminate it from the SA such as the acceptance criterion and the archive maintenance mechanism. In "Performance Evaluation Criteria" we elaborate the performance evaluation metrics used in this study to compare different Pareto fronts in a quantitative manner.

## CNNs—an overview

Neural Networks (NNs) receive an input and transform it through a series of hidden layers each of which is made up of a set of neurons that receives some inputs, performs a dot product followed with a non-linearity. In any layer, each neuron is fully connected to all neurons in the previous layer, that is why Regular NNs don't scale well to inputs in the form of images. Similarly, CNNs are made up of neurons that have learnable weights and biases, and the whole network still expresses a single differentiable score function. In general, CNNs assume that the inputs are images, and they constrain the architecture in a more sensible way. There are three types of layers in CNNs: Convolution Layer, Pooling Layer and Fully Connected Layer.

A convolution layer is a fundamental component of a CNN architecture that performs feature extraction through a combination of linear and nonlinear operations, that is, convolution operation and activation function. Convolution is specialized type of linear operation used for feature extraction with the help of a small array of numbers called filter or kernel navigating over the input tensor. At each location of the input tensor, an element-wise product between the elements of the filter and the input is applied. These products are then summed to obtain the output value in the corresponding position of the output tensor which is called a *feature map*. This operation is repeated using multiple kernels resulting in multiple number of feature maps at a single layer. Figure 1 illustrates a CNN with two convolution layers each of which contains different number of filters with differing size. In the figure, images of size $96 \times 96 \times 3$ are fed into the CNN where the first two dimensions denote the width and height of the image, and the third dimension denotes the number of channels which is 3 for a color image. As can be seen in Fig. 2 which illustrates the convolution process, the number of channels (depth) of the filters always equals to the number of channels of the input image, but the size and the number of those filters are among the hyper-parameters whose values should be selected carefully. The convolution process can be speed up by using dimension reduction adjusting the stride value. If the stride value takes a value other than 1, then the filter skips the input by this amount. If input size reduction is not desired, padding is used. In the pooling layer, size reduction is done in the same way as in the convolution layer with one difference is that the number of channels remains unchanged. There are two main types of
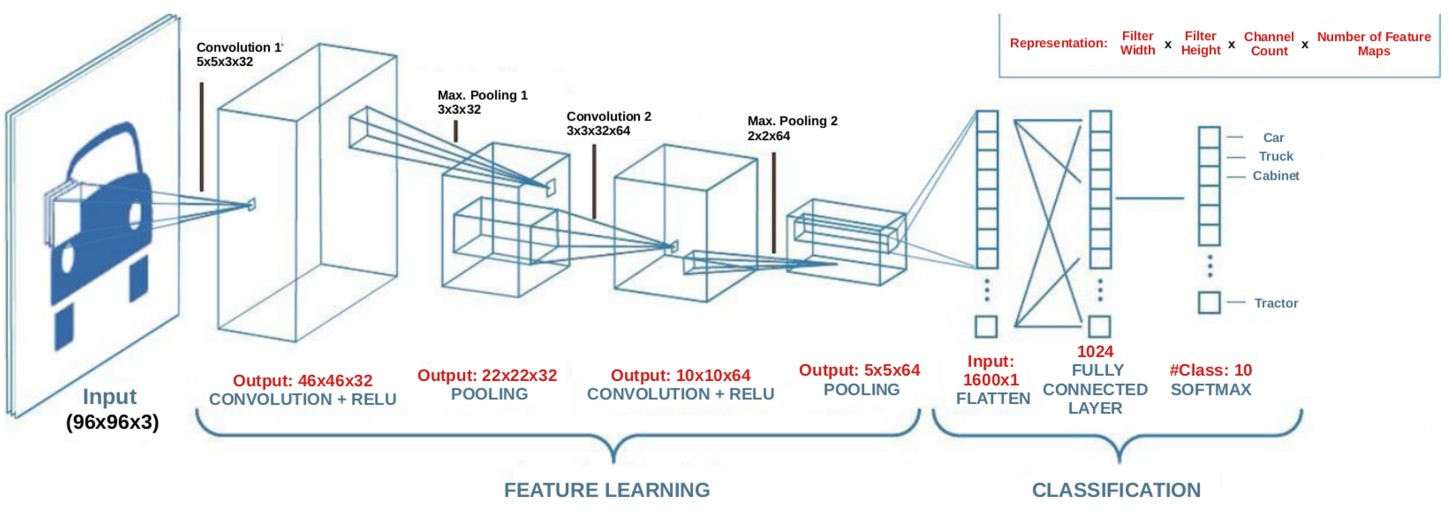

**Figure 1 An example CNN architecture.**

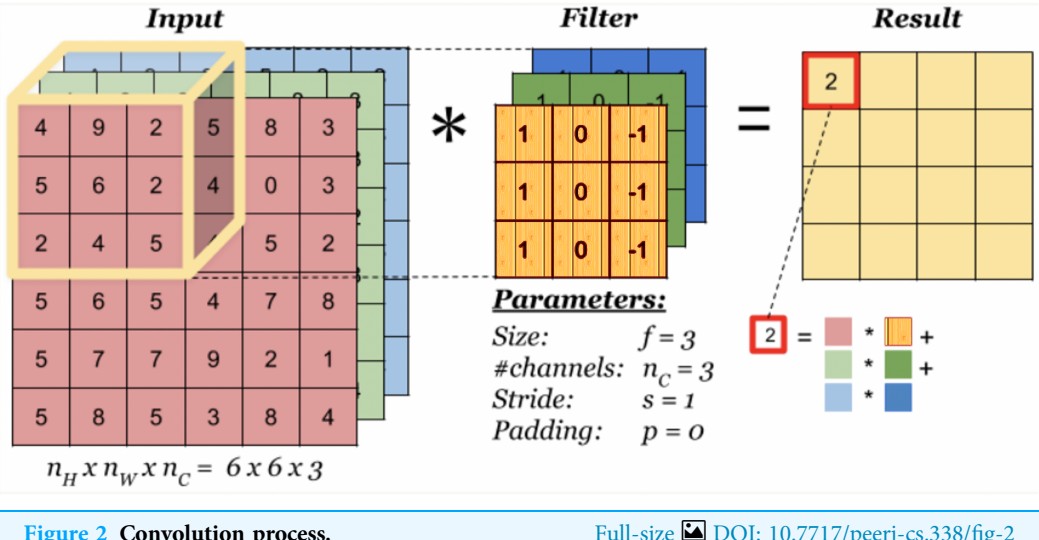

**Figure 2 Convolution process.**

pooling methods: max and average pooling. Pooling type, the size and the stride of the filter are among the hyper-parameters in a given network. In the example CNN in Fig. 1, the first convolution layer includes 32 filters each with a size of $5 \times 5 \times 5$, and after the first convolution operation, linear structure is transformed into a non-linear structure using ReLU activation function. After the pooling layer, input weight and height is reduced to $22 \times 22$, but the depth is increased to 32. After the second convolution and pooling operations, input width and height is reduced even more. Strive method is proposed as an alternative to the pooling method (*Springenberg et al., 2014*). A strive layer is a kind of convolution layer with $3 \times 3$ or $2 \times 2$ filter sizes with a stride of 2. Filters in this layer do not have weights to learn, because, only size reduction is applied. After the convolution and pooling operations, a flattening operation is applied to transform all the feature maps in the final layer into a one-dimensional vector which is then fed to the fully connected

layer where classification results are obtained. There might be multiple fully connected layers in a CNN; however, the output size of the last fully connected layer should match the number of classes in the problem. For a given input, a classification score is computed for each class and the input is assigned to the class with the highest score. Based on the prediction accuracy, an error is computed which is then used to update the weights in the network. This process is repeated until a desired error rate is achieved. For a detailed review on the topic please refer to (*Goodfellow, Bengio & Courville, 2016*).

Hyper-parameters like the total number of layers, the number and size of the filters at each layer along with filter related parameters like stride and padding define a CNN configuration or an architecture. On the other hand, total number of weights in all of the layers of a CNN defines the size or the number of parameters of that network. The size of a CNN is calculated differently for convolution and fully connected layers. The number of parameters in a convolution layer equals to $((d \times m \times n) + 1) \times k$, where d is the number of input feature maps, and m and n are the filter width and height, respectively (1 is added because of the bias term for each filter), and k is the number of output feature maps. The number of parameters in a fully connected equals to $(c + 1) \times p$, where c is the number of input nodes and p is the number of output nodes.

## SA algorithm

Simulated Annealing (SA) uses an adaptation of the Metropolis algorithm to accept non-improving moves based on a probability (*Kirkpatrick, Gelatt & Vecchi, 1983*). In a single-objective SA, a new solution, $X'$, is selected within the neighborhood of current solution $X$, where the neighborhood of $X$ is defined as all the solutions that can be reached by a single move from X. The moves that can be performed in a SA are defined based on the representation used to encode a feasible solution. Solution representation, on the other hand, is highly dependent on the problem domain. If the objective function value of $X'$ is smaller than $X$ (for a minimization problem), then $X'$ is accepted. If $X'$ is worse than $X$, then it is accepted with a probability, $p_{acc}$, which is calculated based on the worsening amount and the current temperature of the system as, $p_{acc} = min\{1, exp(-\Delta F/T_{cur})\}$ where $\Delta F$ is the worsening amount in the objective function and $T_{cur}$ is the current temperature. If the temperature is high, then the probability of accepting a worsening move would be higher than the probability with a lower temperature. In general, the system is initiated with a high temperature to allow exploration during initial steps. As shown in the SA pseudo-code below, after a certain number of solutions are visited at the current temperature level which is defined by the parameter *nbr_inner_iter*, the temperature is lowered gradually according to a predefined annealing scheme. Geometric cooling that uses a decay factor smaller than 1 is the most widely used cooling scheme in SA. At each outer iteration, $T_{cur}$ is reduced to ensure the convergence of the algorithm, because the probability of accepting worsening moves drops as $T_{cur}$ reduces. The total number of solutions visited during a run of a SA is defined as *nbr_out_iter* × *nbr_inner_iter*. However, the number of outer iterations is not always fixed before the start of the algorithm. A minimum temperature level, or a minimum objective function value can be defined as the stopping criterion. Initial

---

**Algorithm 1 SA**

$init\_sa\_params(T_{init}, nbr\_out\_iter, nbr\_inner\_iter)$
$T_{cur} \leftarrow T_{init}$
for $counter \leftarrow 0$ to $nbr\_out\_iter$ do
  for $inner\_counter \leftarrow 0$ to $nbr\_inner\_iter$ do
    $X' \leftarrow SelectFromNeighbor(X)$
    if $(\Delta F = F(X') - F(X)) \leq 0$ then
      $X \leftarrow X'$
    else if $p_{rnd} \leq \exp(-\Delta F / T_{cur})$ then
    $X \leftarrow X'$
    else
    continue with $X$
    end
  end
  $T_{cur} \leftarrow T_{cur}*cool\_ratio$
end

---

temperature, $T_{init}$, cooling rate (if geometric cooling is employed) and the final temperature, $T_{final}$ are important parameters that should be selected carefully.

In this study, the SA algorithm considers only one objective which is the error rate, and it accepts a new solution $X'$ only if it is better than current solution $X$ with respect to this single objective value. However, if $X$ and $X'$ have the same error rate, then the SA selects the one with the smaller number of FLOPs. The composite move used to generate $X'$ is defined in "Local Moves". In order to define $T_{init}$, we use a real time initial temperature selection strategy (*Smith, Everson & Fieldsend, 2004*). In this strategy, $p_{acc}$ is not calculated using the formula $p_{acc} = min\{1, exp(-\Delta F/T_{cur})\}$. Instead, a fixed initial probability value which is recommended as 0.5 is defined for accepting the worsening moves. Then, $T_{init}$ is calculated as $-(\Delta F_{ave}/(ln(p_{acc})))$, where $\Delta F_{ave}$ is the average worsening penalty amount which is calculated executing a short "burn-in" period. A similar real time temperature adjustment approach is also used to define $T_{final}$. In this study, the total iteration budget defines the SA stopping criterion, and the number of inner and outer iterations are defined according to this iteration budget and the cooling scheme.

## MOSA algorithm

There are many types of MOSA algorithms in the literature, and basically, there are two main differences between multi-objective and single-objective SA algorithms. The first difference is the design of the acceptance rule, and second one is the maintenance of an external archive of non-dominated solutions which will eventually yield an approximation of the Pareto front. Let $z_k(X)$ for $k \in \{1, 2, .., K\}$ be the value of the $k^{th}$ objective function of a solution $X$. If a new solution $X'$ yields in objective function values that are superior to $X$ in terms of all of the objectives; that is, $\Delta z_k = z_k(X') - z_k(X)$

---

**Algorithm 2  MOSA**

---

$init\_mosa\_params(T_{init}, nbr\_out\_iter, nbr\_inner\_iter)$

$T_{cur} \leftarrow T_{init}$

for $counter \leftarrow 0$ to $nbr\_out\_iter$ do

    for $inner\_counter \leftarrow 0$ to $nbr\_inner\_iter$ do

        $X' \leftarrow SelectFromNeighbor(X)$

        if $X \succ X'$ // current dominates new

          then

            if $p_{rnd} \le \exp(-\Delta F/T_{cur})$ then $X \leftarrow X'$

        else

        if $X' \succ a, a \in A$ then // an archive solution is dominated by new

            $X \leftarrow X'$

            $updateArchive(X')$

        elseif $a \succ X', a \in A$ then

            // an archive solution dominates new

            $a^* \leftarrow selectRandomArchiveSolution()$

            $X \leftarrow select(a^*, X', X)$ // $(a * $ and $X')$ or $(a *$ and $X)$ competes

        else

            //new does not dominate or is not dominated by any archive solution

            $X \leftarrow X'$

            $updateArchive(X')$

        end

    end

    $T_{cur} \leftarrow T_{cur} * cool\_ratio$

end

---

and $\Delta z_k \le 0, k \in \{1, 2\}$ assuming the two objectives are to be minimized, then it is always accepted. Otherwise, probabilistic acceptance rules as discussed in "SA Algorithm". are used, but before this, multiple objective values should be combined into a single objective value. There are several approaches to combine all objective function values into a single value. (*Ulungu et al., 1999*) use a criterion scalarizing function which takes the weighted sum of the objective functions. *Czyzżak & Jaszkiewicz (1998)* use a diversified set of weights to obtain a diverse set of solutions in the final front in their SA-based algorithm which they call Pareto Simulated Annealing (PSA). *Suppapitnarm et al. (2000)* propose another MOSA, which they call SMOSA that suggests maintaining different temperatures, one for each objective. In the study a "return-to-base" strategy which restarts the search from a random archive solution is introduced. In dominance-based acceptance rules, $X$ and $X'$ are compared with respect to the dominance relation. $X$ is said to dominate $X'$ which is denoted as $X \succ X'$, if it is better than $X'$ in at least one objective and it is not worse than $X'$ in all of the other objectives. If at any iteration, $X \succ X'$ then, $X'$ is accepted with a probability which is also computed according to the domination status of the solutions. *Suman (2004)* proposes a Pareto Domination-Based MOSA which uses the

current domination status of a solution to compute its acceptance probability. The acceptance probability of a solution is determined by the number of solutions in the current front dominating it. *Bandyopadhyay et al. (2008)* proposes Archive-based Multi-objective Simulated Annealing (AMOSA) which again uses a dominance-based acceptance rule. A detailed survey on single-objective and multi-objective SA can be found in *Suman & Kumar (2006)*.

The MOSA algorithm proposed in this study iterates in a similar fashion as the single-objective SA algorithm. However, there are now two objective values to take into consideration while moving from $X$ to $X'$, where the first objective being the number of FLOPs required by the network, and the second being the error rate achieved by the network. Criterion scalarizing rules can be used to combine the two objective values into one, but this method requires both objective values to be in the same scale and proper weights to be defined for each objective. On the other hand, probability scalarizing rules calculate the acceptance probability of $X'$ for each objective individually, then a decision rule is applied to aggregate these probabilities such as taking the minimum, maximum or the product of these probabilities. As different probabilities are evaluated for each objective, a different temperature should be maintained for each objective in this approach. Due to the difficulties mentioned above, we adopt the Smith's Pareto dominance rule (*Smith et al., 2008*) in our MOSA algorithm. Starting from the first iteration, all non-dominated solutions encountered during the search are collected in an external archive, $A$, which is updated whenever a new solution is accepted. The state of $A$ is expected to improve as the algorithm iterates, and archive solutions in the final iteration form the Pareto front which is presented to the decision maker. As shown in the MOSA pseudo-code below, as the algorithm iterates from $X$ to $X'$, if $X \succ X'$ ($X$ dominates $X'$), then $X'$ is accepted based on Smith's Pareto dominance rule which calculates the acceptance probability of a given solution by comparing it with the current archive which contains potentially Pareto-optimal set of solutions. Smith's rule uses a difference in the energy level, $\Delta F$, to compare two solutions with respect to all objectives as follows: Let A denote the current potentially Pareto optimal set and $\tilde{A}$ denote the union of this set and two competing solutions, $\tilde{A} = A \cup X \cup X'$, then $\Delta F = \{1/|\tilde{A}|\} \cdot \{|F(X')| - |F(X)|\}$, where $F(X)$ denotes the solutions in A that dominate $X$ plus 1. The probability of accepting $X'$ is then computed as $p_{acc} = exp(-\Delta F/T_{cur})$, where the only difference with a single-objective version is the calculation of $\Delta F$. In this approach, only one temperature is maintained regardless of the number of objectives. In a multi-objective version of SA algorithm, archive maintenance related rules should also be defined. If $X \nsucc X'$ ($X$ does not dominate $X'$), then $X'$ becomes a candidate for being a member of the archive. This is determined as follows: If $X'$ dominates any solution in A, then $X'$ is accepted and the archive is updated by inserting $X'$ and removing all the solutions dominated by it. If $X \nsucc X'$ but there is a solution $a \in A$ dominating $X'$, then no archive update is performed and the current iteration is completed either by accepting $X'$, $a$ or $X$ as the current solution. If $X' \succ X$, then $a$ and $X'$ compete for being selected as the current solution according to the probability calculated with the same rule described above. Here, $a$ represents a random

archive solution dominating $X'$. Continuing the search process with a previously visited archive solution is known as *return-to-base* strategy. If $X \not\succ X'$ and $X' \not\succ X$, then first these two solutions compete, and then the winning solution competes with $a$ according to the same probabilistic acceptance rule. If $X \not\succ X'$ and there is no solution $a \in A$ that dominates $X'$, and $X'$ does not dominate any solution $a \in A$, then $X'$ is accepted and inserted into the archive. As the archive is updated by removing all dominated solutions, the size of the archive does not grow very large. Selection of other MOSA parameters such as $T_{init}$, $T_{final}$, cooling rate and also the number of inner and outer iterations are given in detail in Section "MOSA parameter tuning".

## Performance evaluation criteria

In multi-objective optimization problems, final fronts are evaluated according to two performance criteria: (i) Closeness to the Pareto-optimal front and, (ii) Diversity of the solutions along the front (*Zitzler, Deb & Thiele, 2000*; *Deb, 2001*). Generational Distance (GD) is the most-widely used metric to examine the closeness of the final front to Pareto-optimal front (*Van Veldhuizen & Lamont, 2000*). In order to measure the diversity which is comprised of two components, namely distribution and spread, two metrics, spacing and maximum spread are used. Spacing evaluates the relative distance among the solutions, whereas spread evaluates the range of the objective function values.

GD metric requires a Pareto-optimal front in order to compare any two fronts. Since this optimal front is not known in advance, it is approximated by an aggregate front, $A^*$, which is formed by combining all solutions in the two fronts. Then, the amount of improvement achieved in each front is measured with respect to this aggregate front. GD for a given front is calculated as given in Eq. (1), where $d_i$ is the Euclidean distance between the solution $i \in A$ and the closest solution $k \in A^*$. In Eq. (2), $P^{max}$ and $E^{max}$ denote the maximum objective values, whereas $P^*$ and $E^*$ denote the minimum objective function values observed in $A^*$. For this GD metric, smaller the better.

$$GD(A) = \frac{\sqrt{\sum_{i \in A} d_i^2}}{|A|} \tag{1}$$

$$d_i = \min_{k \in A^*} \sqrt{\frac{1}{2}\left(\left(\frac{P_i - P_k}{P^{max} - P^*}\right)^2 + \left(\frac{E_i - E_k}{E^{max} - E^*}\right)^2\right)} \tag{2}$$

Zitzler's spread metric which is computed as in Eq. (3) is used to measure the extent of the fronts. This metric simply calculates Euclidean distance between the extreme solutions in a given front. If a front $A$ includes all the extreme solutions in $A^*$, then this front takes a value of 1 according to this metric.

$$S(A) = \sqrt{\frac{1}{2}\left(\left(\frac{\max_{i \in A} P_i - \min_{i \in A} P_i}{P^{max} - P^*}\right)^2 + \left(\frac{\max_{i \in A} E_i - \min_{i \in A} E_i}{E^{max} - E^*}\right)^2\right)} \tag{3}$$

Spacing metric (*Schott, 1995*) is used to measure the diversity along a given front. For each solution, distance to its closest neighbor is calculated as shown in Eq. (4). Then, the spacing metric is calculated as the standard deviation of these distances as shown in Eq. (5), where $\bar{d}$ is the average distance value. If the distance between the closest solutions are distributed equally, then the value of this metric approaches to zero which is the desired condition.

$$d_i = \min_{k \in A \wedge k \neq i} \left( \frac{|P_i - P_k|}{\max\limits_{j \in A} P_j - \min\limits_{j \in A} P_j} + \frac{|E_i - E_k|}{\max\limits_{j \in A} E_j - \min\limits_{j \in A} E_j} \right), \text{ for } i \in A \tag{4}$$

$$Sp(A) = \sqrt{\frac{1}{|A|} \sum_{i \in A} \left( d_i - \bar{d} \right)^2} \tag{5}$$

# RESULTS

## Implementation details

### Solution representation and the search space

Solutions can be represented with either block (*Ma et al., 2020*) or layer structure (*Yamasaki, Honma & Aizawa, 2017*; *Sun et al., 2019b*). In this study, we adopted the block structure with variable length representation; where the solutions are allowed to expand or shrink dynamically. A new solution is generated by repeating a certain number of convolution and fully connected blocks. A convolution block is composed of a convolution layer (CONV), activation function (ACT), batch normalization (BN), subsampling method (SUBS) and a dropout function (DROP). A fully connected block is composed of a fully connected layer (FC), activation function, batch normalization and a dropout function. This structure can be represented as: $[(CONV \rightarrow ACT \rightarrow BN) * N_C \rightarrow (SUBS \rightarrow DROP)] * N_{CB} \rightarrow [(FC \rightarrow ACT \rightarrow BN \rightarrow DROP)] * N_{FB}$, where #*Conv* denotes the number of number of convolution layers in a convolution block, $N_{CB}$ denotes the number of convolution blocks, and $N_{FB}$ denotes and the number of fully connected blocks. We adopted the same dictionary structure given in (*Ma et al., 2020*) to represent a solution which is given below:

$$\left\{ \left\{ Conv : \{ks, kc, p, s, a\}_{N=1}^{\#Conv}, Pool : \{ks_p, s_p, pt, d_p\} \right\}_{M=1}^{N_{CB}} + \{u_f, d_f, a_f\}_{K=1}^{N_{FB}} \right\}$$

We adopted the same hyper-parameters as in (*Gülcü & Kuş, 2020*), and the full name of the hyper-parameters whose abbreviations are given in the dictionary representation are given in Table 1 along with their type and value ranges. An example solution that uses the dictionary representation is given below (please refer to Table 1 for the abbreviations):

{{$N_{CB}$: 2, $N_{FB}$: 1},
{"Conv_Block_1": {"Conv" : {#Conv: 2, ks: 5, kc: 32, p: SAME, s : 1, a: relu}, "Pool" : {$ks_p$: 3, $s_p$: 2, pt: MAX, $d_p$ 0.2 }}, "Conv_Block_2" : {"Conv" : {#Conv: 3, ks: 3, kc: 64, p: SAME, s : 1, a: relu}, "Pool" : {$ks_p$: 3, $s_p$: 2, pt: MAX, $d_p$: 0.4}}} + "Fully_Block_1" : {$u_f$: 128, $d_f$: 0.5, $a_f$: relu}}.

**Table 1 Hyper-parameters to be optimized, their value types and ranges.**

|  | Hyper-parameter | Abbreviation | Value types | Value ranges |
|---|---|---|---|---|
| Convolution (Conv) | Filter size | ks | Numeric | 3, 5, 7 |
|  | Filter count | kc | Numeric | 32, 64, 96, 128, 160, 192, 224, 256 |
|  | Padding | p | Categorical | SAME** |
|  | Stride | s | Numeric | 1** |
|  | Activation function | f | Categorical | ReLU, Leaky-ReLU, Elu |
|  | Subsampling method | – | Categorical | Pooling, Strive |
| Pooling (Pool)* | Filter Size | $ks_P$ | Numeric | 2, 3 |
|  | Stride | $s_P$ | Numeric | 2** |
|  | Type | pt | Categorical | MAX, AVG |
|  | Dropout rate | $d_P$ | Numeric | 0.3, 0.4, 0.5 |
| Strive* | Strive Filter Size | $ks_s$ | Numeric | 2, 3 |
|  | Padding | $p_s$ | Categorical | Valid** |
|  | Stride | $s_s$ | Numeric | 2** |
| Fully Connected | Number of units | $u_f$ | Numeric | 128, 256, 512 |
|  | Dropout rate | $d_f$ | Numeric | 0.3, 0.4, 0.5 |
|  | Activation function | $a_f$ | Categorical | ReLU, Leaky-ReLU, Elu |
|  | #Convolutional layers | #Conv | Numeric | 2, 3, 4 |
|  | #Convolutional blocks | $N_{CB}$ | Numeric | 2, 4 |
|  | #Fully connected layers | $N_{FB}$ | Numeric | 0, 2 |

**Notes:**
  * Conditioned on subsampling method.
  ** Fixed values.

## Local moves

MOSA algorithm starts by taking VGGNet (*Simonyan & Zisserman, 2015*) as the initial solution, and at any iteration, a new solution is created from the current solution by modifying the convolution and fully connected block hyper-parameters as follows:

**Step1:** add a new convolution block with a probability p which takes a value of 0.0625 initially, and increases by 1.4 times at every 50 iterations. A newly added block inherits all the hyper-parameters from the preceding block.

**Step 2:** select the subsampling method, pooling or strive, with equal probabilities.

**Step 3:** start from the first convolution block, modify each block as follows:

- if *#Conv* < maximum allowed layer count, then, add a new convolution layer with a *p* of 0.8; otherwise, delete the last convolution layer with a *p* of 0.2
- modify the convolution layer hyper-parameters with a *p* of 0.5. If it is decided to be modified, then only one hyper-parameter which is selected randomly is modified. Since the same layer hyper-parameters are used within the same block (as in the VGGNet architecture), this selection affects all the layers in that block.

**Step 4:** add a new fully connected block in a similar fashion to adding a new convolution block.

**Step 5:** start from the first fully connected block, modify only one random hyper-parameter with a $p$ of 0.5 as in the convolution block.

We have observed during the preliminary experiments that the approach of adding a new convolution block with an initial probability of 0.0625, and increasing this probability by 1.4 times at every 50 iterations, enabled the search reach 4-block networks around iteration number 325. With this approach, the probability of adding a new block is increased to 0.24 at the end of the 200th iteration; and at the end of 300th iteration, it became 0.46 allowing the network to grow with roughly 50% probability. If we had kept this probability constant, 4-block networks would have been explored for only a few iterations towards the end of the search process.

## Experimental setup
### MOSA parameter tuning
In our study, the MOSA algorithm adopts a real time initial temperature selection strategy in which the worsening moves are accepted with an initial probability, $p_{acc}$, of 0.5. $T_{init}$ is then calculated as $T_{init} = -(\Delta F_{ave}/(ln(p_{acc})))$. Average initial worsening penalty amount, $\Delta F_{ave}$, is approximated executing a short "burn-in" period in which worsening moves as well as improving ones are all accepted. We run a burn-in period with 100 iterations and 39 of them resulted in worsening solutions. In Table 2, some information regarding only a few of those iterations are given. In the table, $F(X)$ denotes the solutions in A that dominate X plus 1, $|A|$ denotes the size of the archive, $F(X') - F(X)$ denotes the worsening amount in terms of domination count and $\Delta F$ denotes the objective value calculated as $\Delta F = \{1/|\tilde{A}|\} \cdot \{|F(X')| - |F(X)|\}$, where $|\tilde{A}|$ equals $|A| + 2$. The average $|A|$ value obtained in this burn-in period is used to calculate $\Delta F_{ave}$. This period results in a $\Delta F_{ave}$ value of 0.40 which also gives a $T_{init}$ value of 0.577. We adopt a similar temperature adjustment approach to determine $T_{final}$ value, where in this case we assumed a $F(X') - F(X)$ value of at most 1 to be accepted with the same probability. In order to calculate $T_{final}$, final front size needs to be approximated, and a value of 10 for the final front size seemed to be reasonable according to preliminary experiments, and $T_{final}$ is calculated as 0.12.

In this study, the MOSA algorithm is allowed to run for a number of iterations defined by the iteration budget of 500, which means that at most 500 solutions are created during a single run. The amount of this total iteration budget to be allocated in outer and inner iterations is determined by the cooling rate parameter. We tested for several cooling rate values keeping the total number of iterations at 250 due to large computational times. Table 3 shows the number of outer and inner iterations calculated for different cooling rate values under the same budget and temperature values. Based on these iteration numbers, we selected cooling rate values of 0.95, 0.90 and 0.85 for the parameter tuning process. We allowed the MOSA algorithm run for 3 times for each of those cooling rate values using different random number seeds. The fronts obtained under each cooling rate value for three runs are given in Fig. 3. As can be seen in the figure, three Pareto fronts are obtained for each cooling rate value. Although one can notice that there is no difference among the fronts formed with different cooling rate values visually, we applied

**Table 2 Worsening moves encountered during burn-in period.**

| Iteration no | $F(X)$ | $F(X')$ | $|A|$ | $F(X')-F(X)$ | $\Delta F$ |
|---|---|---|---|---|---|
| 3 | 1 | 4 | 3 | 3 | 0.600 |
| 5 | 5 | 4 | 4 | 4 | 0.667 |
| 9 | 1 | 2 | 5 | 1 | 0.143 |
| 11 | 1 | 6 | 6 | 5 | 0.625 |
| 13 | 1 | 7 | 7 | 6 | 0.667 |
| ⋮ | ⋮ | ⋮ | ⋮ | ⋮ | ⋮ |
| Average | 2.69 | 5.94 | 6.02 | 3.25 | 0.40 |

**Table 3 The number of outer and inner iterations calculated for different cooling rate values under the same budget and temperature values.**

| Iteration budget | $T_{init}$ | $T_{final}$ | Cooling rate | #Outer iterations | #Inner iterations |
|---|---|---|---|---|---|
| 250 | 0.577 | 0.12 | 0.99 | 156.2 | 1.6 |
| 250 | 0.577 | 0.12 | 0.95 | 30.6 | 8.1 |
| 250 | 0.577 | 0.12 | 0.9 | 14.9 | 16.7 |
| 250 | 0.577 | 0.12 | 0.85 | 9.6 | 25.8 |
| 250 | 0.577 | 0.12 | 0.8 | 7.0 | 35.5 |

Kruskal–Wallis $h$-test to determine whether there are significant differences. This test is applied separately for GD, Spread and the front size metrics, but none of those metrics proved a significant difference among the fronts. Therefore, we selected a cooling rate of 0.85 arbitrarily.

### Cifar10 dataset

In this study, CIFAR-10 dataset (*Krizhevsky & Hinton, 2009*) which is the most widely-used natural object classification benchmark dataset used in HPO and NAS studies is selected as the benchmark dataset. Considering the computational cost and the hardware requirement to run the experiments, the selected dataset should be simple, yet complex enough to reveal the differences among different methods or the network configurations. Many studies consider this dataset as the only benchmark dataset (*Lu et al., 2019a*; *Wang et al., 2019*; *Elsken, Metzen & Hutter, 2018*) due to the fact that it is simpler than very large-scale ImageNet dataset (*Russakovsky et al., 2015*), but still difficult enough to be used to evaluate the performance of different approaches.

   CIFAR-10 dataset consists of 60,000 32 × 32 color images in 10 classes, and each class contains 6,000 images. The whole dataset is divided into training and test datasets of 50,000 and 10,000 images, respectively. In most of the studies, CIFAR-10 original training set is split into two sets (80–20%) to create training and validation sets which are used during the search process. We follow a similar approach with only difference is that, only half of the original training dataset is used during a MOSA search process. In order to speed up the search process, a reduced sample of 50% of the original training samples are

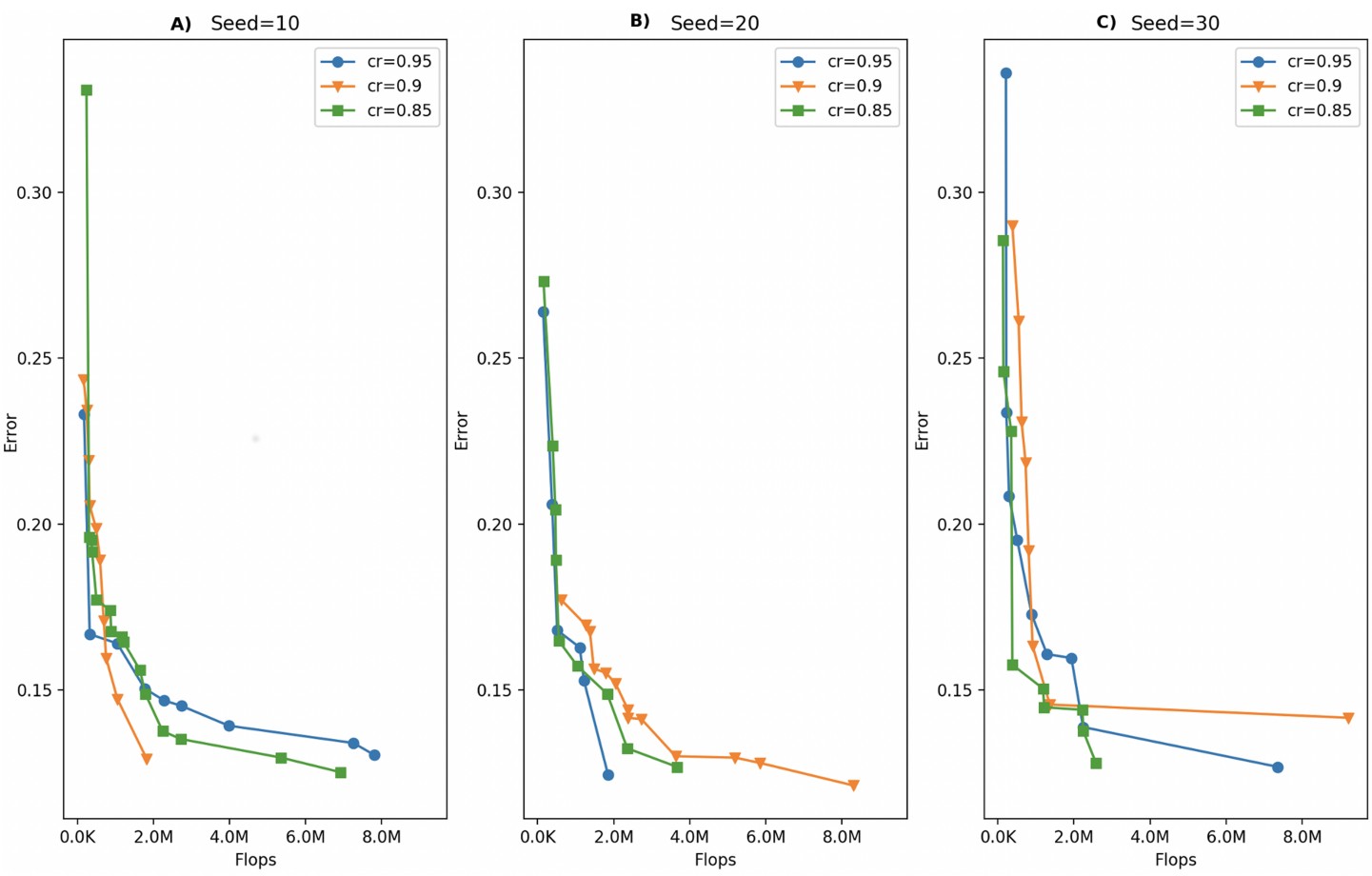

**Figure 3 Comparison of MOSA fronts under different cooling rates with (A) random seed: 10, (B) random seed: 20, (C) random seed: 30.**

selected randomly; and 10% of this reduced sample is used as the reduced validation set. Although this approach might have some negative effects on the performance evaluation of a given configuration; we believe this effect is minimal due to the fact that the aim of the search process is not to accurately measure the error rate of a configuration, but to perform a fair comparison to discriminate good and bad configurations. The original CIFAR-10 test set is never used during the MOSA search process, and it is only used to obtain the actual test accuracy values of the selected configurations on the final trade-off front. Other image classification datasets such as MNIST (*LeCun et al., 1998*), FashionMNIST (*Xiao, Rasul & Vollgraf, 2017*) and EMNIST-Balanced (*Cohen et al., 2017*) are being criticized for having reached their limits and failing to reveal the differences between the algorithms (*Lu et al., 2019a*).

### Training and evaluation during the search process
During the search process, the classification performance of a generated network configuration is approximated by following the early stopping approach. Early stopping is used as a popular method to prevent over-fitting in classical machine learning; however, in

the context of AutoML, and in particular for HPO, it is usually used to cut down the training time for unpromising configurations. Based on the evaluations on the validation set, if a poor early-stage performance is observed, then the training process is terminated, and the search moves to a new configuration. This approach not only cuts down total running time of the HPO method, but also introduces noise and bias to the estimation, as some configurations with bad early-stage performance may eventually turn out to be good after sufficient training (*Yao et al., 2019*). At each MOSA iteration, a newly generated configuration is trained using the training split of the original training set and it is evaluated on the validation set which is the test split of the original training set. Xavier weight initializer and Adam optimizer with a learning rate of 0.001 is used, and the batch size is selected as 32. For a given configuration, if the best loss achieved on the validation set is not improved after three consecutive epochs, then the training process is terminated. A configuration is allowed to be trained at most 100 epochs. In a MOSA run with 500 iterations, we observe that the average number of epochs is 23.166. This epoch count seems to be reasonable considering the epoch count used in similar studies: minimum of 36 epochs in (*Lu et al., 2019a*) and 20 epochs in *Elsken, Metzen & Hutter (2018)*. All experiments are performed on a single Nvidia 2080 Ti GPU using Keras (*Chollet et al., 2015*), and the code and the raw evaluation results are available at https://github.com/zekikus/MOSA-cnn-hyperparams-optimization.

### Training after the search process

The MOSA algorithm is run for 500 iterations, and each run is repeated for three times using different random seeds. From each of three trade-off fronts, some non-dominated configurations are selected and then trained for longer epochs on the original CIFAR-10 training dataset in order to measure their actual classification performance. From each front, three configurations with the lowest error rates are selected and each solution is trained for 400 epochs with a batch size of 128 using standard stochastic gradient descent (SGD) back-propagation algorithm with the following parameter values: learning rate = 0.08, decay = 5E−4 and momentum = 0.9 (default values in Keras). As mentioned earlier, the original test set is never used during training; it is only used at this stage to test the performance of the selected networks. To improve the test performance, we only utilized an online augmentation routine that is used in many peer studies. This process which is also called augmentation on the fly is especially preferred for larger datasets where an increase in size cannot be afforded. Sequential transformations of padding, random crop and horizontal flip are applied on the mini-batches during training (*He et al., 2016*; *Huang et al., 2017*). In some studies, training performance is further improved by some additional operations. For example, (*Lu et al., 2019b*) appends an auxiliary head classifier to the architecture, but we did not follow these approaches that require manual intervention after the search process.

### Results analysis

We first present the objective space distribution of all configurations obtained during each independent run of the MOSA. In order to show the search ability of the proposed

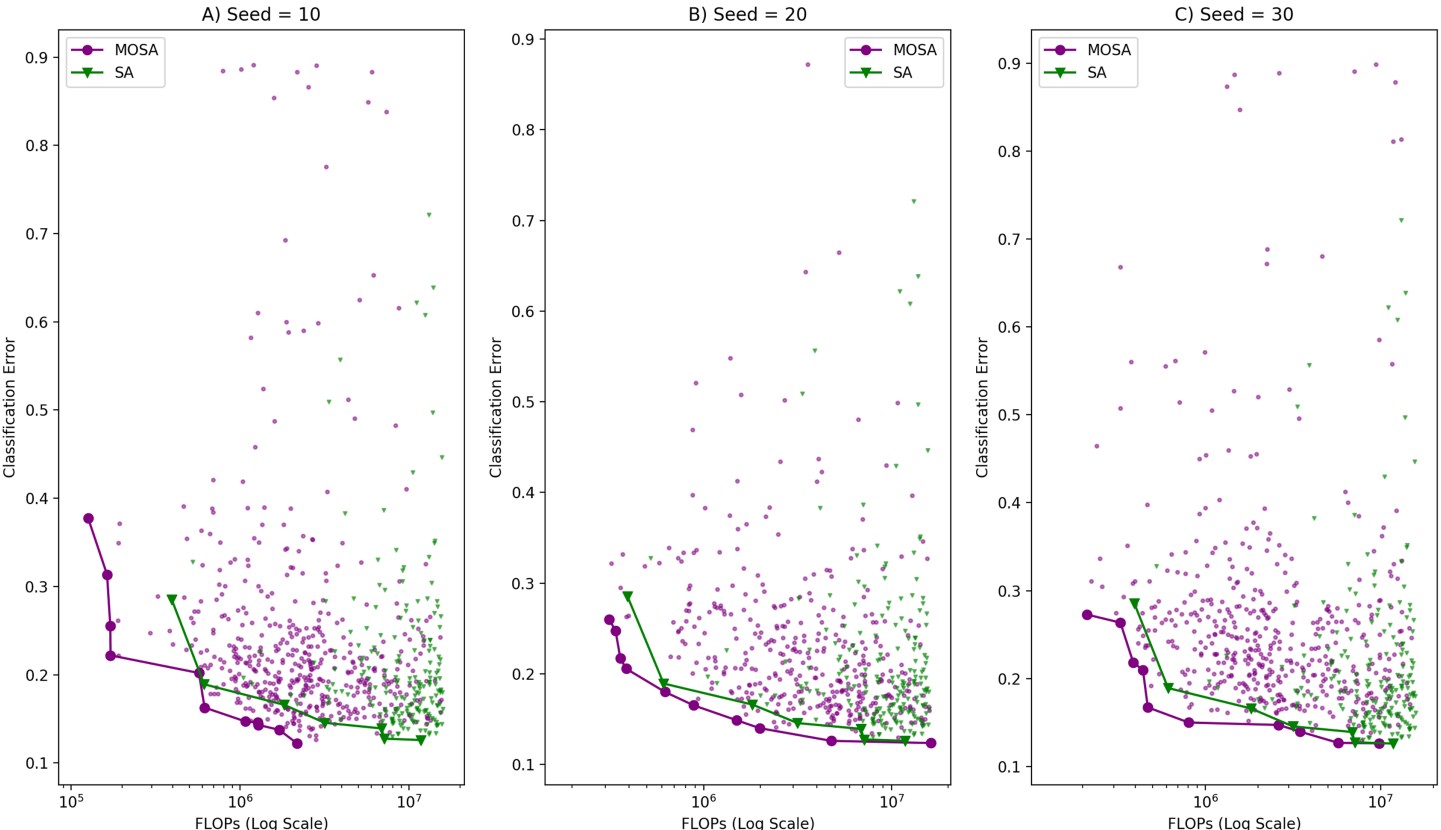

**Figure 4** Comparison of MOSA and SA search ability in terms of objective space distribution and the Pareto fronts with (A) random seed: 10, (B) random seed: 20, (C) random seed: 30.

algorithm, the MOSA search space is compared to the search space generated by the single-objective SA method also to the search space generated by Random Search (RS) method in which all solutions are accepted regardless of their objective values. Final fronts are also compared in terms of the front metrics given in "Performance Evaluation Criteria". Then, some solutions in the final front are selected to be trained for longer epochs in order to perform post search analysis. The configurations generated by the MOSA are compared to both human designed state-of-the-art configurations and the configurations generated by other search methods like evolutionary algorithms in terms of the test error, the number of FLOPs, the number of parameters and also the search cost which is reported as GPU-days.

### Fronts analysis

A MOSA search is repeated three times with different initial random seeds, and the objective space distribution of all configurations encountered during each of those search processes are illustrated in Fig. 4. In all of those runs, SA especially focuses on the areas with small classification error but with large number of FLOPs, es expected. In order to show the search ability of the MOSA over the SA, we compare the final fronts generated by each method in terms of closeness to the Pareto-optimal front and the diversity of the

**Table 4 Evaluation of the final fronts with respect to three metrics and the number of solutions.**

|  | MOSA_01 | MOSA_02 | MOSA_03 | SA | RS |
|---|---|---|---|---|---|
| GD | 0.0011 | 0.0629 | 0.0387 | 0.0765 | 0.0286 |
| S (spread) | 0.7127 | 0.7944 | 0.5867 | 0.6681 | 0.4755 |
| Sp (spacing) | 0.0891 | 0.1483 | 0.1384 | 0.1519 | 0.1268 |
| #Solutions | 11 | 10 | 10 | 7 | 10 |

solutions along the fronts. The configurations generated by the SA seem to be much complex than the configurations generated by the MOSA which also takes into account the complexity as the other objective. Due to the large computational time requirement (at least $2 \times$ longer), SA is run only once. As there is no archive mechanism in the SA, the final SA front is formed by selecting all non-dominated configurations encountered during a single run. The MOSA and the SA fronts are also shown in Fig. 4 along with the objective space distributions of those algorithms. In addition to making comparison using graphical plots, we used the metrics given in "Performance Evaluation Criteria" to compare the fronts generated by each approach in a quantitative manner. These metrics are as follows: Generational Distance (GD), Spread (S) and Spacing (Sp) metrics. The comparison results using these metrics are given in Table 4. In the table, MOSA_01 represents the MOSA front obtained at the first run, and MOSA_02 represents the MOSA front obtained at the second run and so on. GD of each front is calculated with respect to $A^*$ which is formed by combining all four fronts. S of each front is calculated with respect to the extreme points considering again all four fronts. Sp metric is calculated for each front separately, because it measures within front distribution quality of a given front.

The search ability of the proposed method is also validated by comparing it to the RS method. The same comparisons performed between the MOSA and the SA are applied to compare the MOSA and the RS methods. When the MOSA and the RS solutions are combined to create $A^*$, none of the RS front solutions take place in $A^*$ as in the case of SA. $A^*$ is composed of only the solutions coming from the MOSA which means that GD calculations are made against the same reference front; therefore we decided to present all these comparison results on the same table, Table 4. Figure 5 illustrates the objective space distribution and the fronts obtained by the MOSA and the RS methods.

### After search analysis

In order to measure the actual classification performance of the configurations generated by the MOSA method, nine configurations are selected and subjected to longer training using the original CIFAR-10 training dataset as detailed in "Training After the Search Process". Among these nine configurations, the networks with the lowest error rates are selected for comparison with the networks reported in the literature. As the MOSA algorithm considers two objectives, namely, error rate and the FLOPs, during the search process, a multi-objective comparison with respect to the front evaluation metrics should be performed for a fair comparison. Unfortunately, most of the studies do not

**Peer**J Computer Science

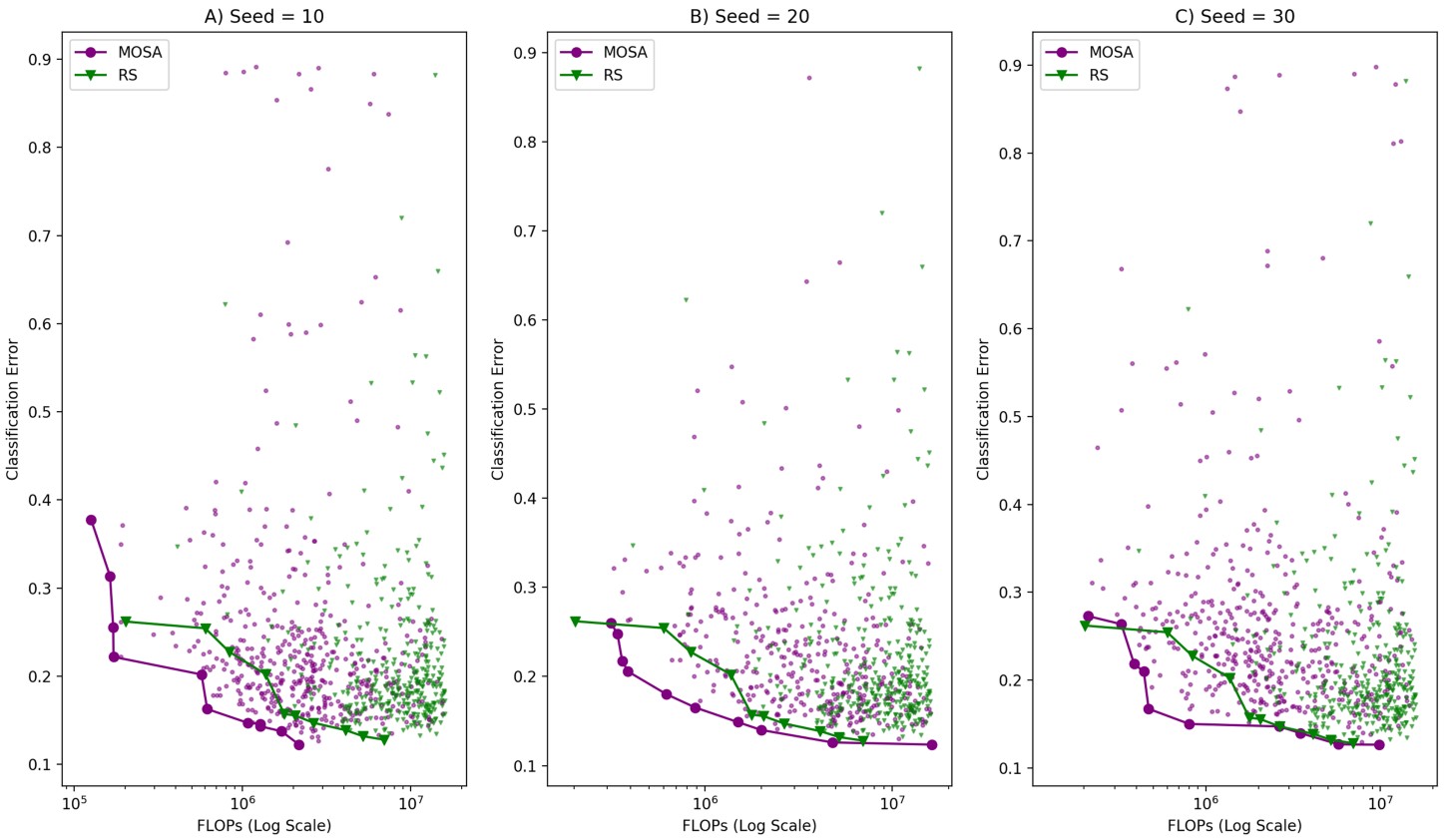

**Figure 5 Visual comparison of MOSA and RS search ability in terms of objective space distribution and the Pareto fronts with (A) random seed: 10, (B) random seed: 20, (C) random seed:30.**

report the objective values of the solutions in the final frontiers, or they consider different objective functions to estimate the complexity of the generated networks. In some studies, computational complexity is measured as computational time in GPU-days. However, a comparison with respect to computational time may not be reliable due to some external factors such as the temperature in the computing environment, even if the same GPU models are used for the experiments. Therefore, the number of FLOPs that a network carries out during a forward pass is the most reliable complexity measure. In Table 5, the final performance of three MOSA configurations in terms of both objectives are presented. In addition, a graphical comparison is given in Fig. 6 where the dominance relation between different configurations can be easily noticed. As in "Fronts Analysis", a comparison to the configurations found with the single objective SA algorithm is also performed and the results are presented in the table in order to show the effect of using a multi-objective solution approach for this bi-objective optimization problem. From the SA trade-off front, three configurations with the lowest error rate are selected for longer training, and the performance of each configuration is reported in the table. The same approach is followed for the RS, as well. Other search generated architectures are included in the second part of the table. In the table, the accuracy, the number of FLOPs and the number of parameters columns all represent the values reported in the

**Table 5 Comparison of MOSA architectures to other search generated architectures.**

| Architecture | Search method | Test accuracy (%) | FLOPs (M) | Parameters (M) |
|---|---|---|---|---|
| MOSA_soln1 | MOSA | 91.97 | 16.2 | 3.329 |
| MOSA_soln2 | MOSA | 91.52 | 1.7 | 0.854 |
| MOSA_soln3 | MOSA | 91.14 | 1.2 | 0.641 |
| SA_soln1 | SA | 92.65 | 6.8 | 5.9 |
| SA_soln2 | SA | 92.85 | 11.7 | 3.572 |
| SA_soln3 | SA | 90.04 | 7.1 | 3.45 |
| RS_soln1 | RS | 88.27 | 4.1 | 2.052 |
| RS_soln2 | RS | 91.25 | 5.1 | 2.603 |
| RS_soln3 | RS | 86.26 | 1.7 | 0.887 |
| NSGA-Net (*Lu et al., 2019b*) | EA | 96.15 | 1290 | 3.3 |
| PPP-Net (*Dong et al., 2018*) | EA | 95.64 | 1364 | 11.39 |
| AmoebaNet-A + cutout (*Real et al., 2019*) | EA | 97.23 | 533 | 3.3 |
| NASNet-A + cutout (*Zoph et al., 2018*) | RL | 97.09 | 532 | 3.2 |

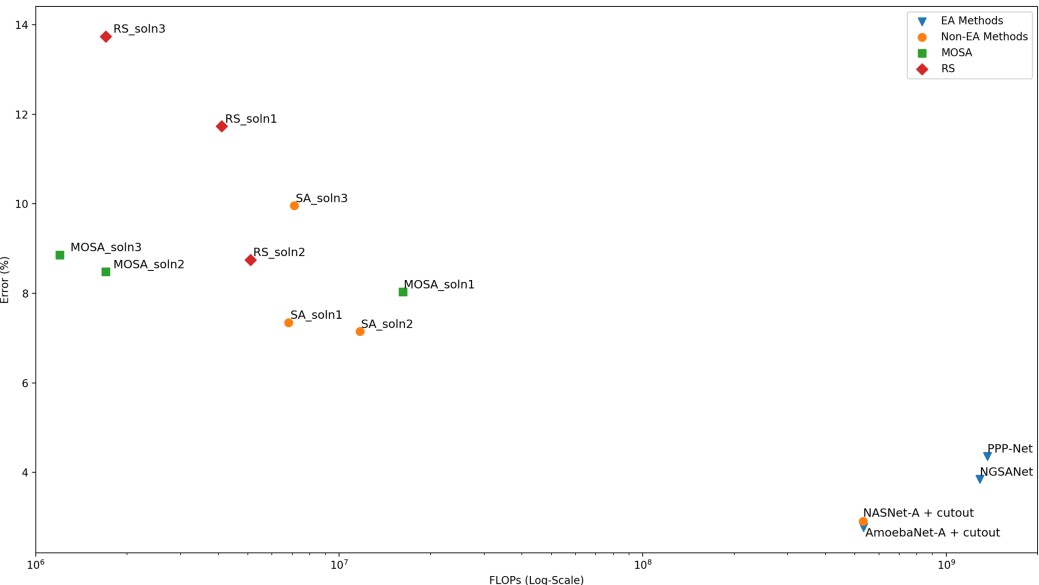

**Figure 6 Comparison of the final performance of the networks in terms of both the error rate and the number of FLOPs.**

original papers. As stated before, we did not consider the search cost in terms of time in the comparisons; however, in order to give an idea about the search cost of the MOSA algorithm, we run one MOSA search on the same GPU as in NSGA-Net (*Lu et al., 2019a*). We observe that MOSA takes 50 hours on a single NVIDIA 1080Ti GPU, which equals to 2.08 GPU-days, whereas it takes 8 GPU-days for NSGA-Net.

A comparison of the MOSA configurations to the human designed state-of-the-art configurations are also performed. In order to be able to make a fair comparison, especially

**Table 6 Comparison of MOSA architectures to human designed state of the art architectures.**

| Architecture | Test accuracy (%) | FLOPs (M) | Parameters (M) |
|---|---|---|---|
| MOSA_soln1 | 91.97 | 16.2 | 3.329 |
| MOSA_soln2 | 91.52 | 1.7 | 0.854 |
| MOSA_soln3 | 91.14 | 1.2 | 0.641 |
| LeNet-5 (*LeCun et al., 1990*) | 71.86 | 0.162 | 0.081 |
| VGGNet-16 (*Simonyan & Zisserman, 2014*) | 86.38 | 67.251 | 33.638 |
| ResNet-50 (*He et al., 2016*) | 76.01 | 47.052 | 23.604 |
| DenseNet-32 (*Huang et al., 2017*) | 89.38 | 2.894 | 0.494 |

in terms of test accuracy, each of these state-of the-art architectures are rebuilt and trained using the same augmentation techniques as in the MOSA training process. The results are presented in Table 6.

## DISCUSSION

The MOSA and the SA algorithms are allowed to run for the same number of iterations. An archive is maintained to collect all non-dominated solutions that have been encountered throughout the search process. The MOSA also incorporates a return-to-base strategy which allows further exploitation of the archive solutions. Moreover, it uses a dominance-based acceptance rule as the decision criterion. CIFAR-10 dataset which is the most widely-used natural object classification benchmark dataset is used to compare the performance of different approaches. We first compare the search ability of the MOSA and the SA algorithms using the trade-off fronts obtained by each method. Objective space distribution of all configurations encountered during a SA search process is used to form the trade-off front consisting of only non-dominated solutions. The MOSA and the SA fronts are then compared with respect to three multi-objective evaluation metrics. According to the results based on generational distance, spread and spacing metrics, the MOSA algorithm is able to generate better fronts than the SA method. When the objective space distribution of the two methods are compared visually, one can see that the single-objective SA focuses on the objective space with small error rates regardless of the number of FLOPs, as expected. On the other hand, the MOSA focuses on the objective space with both small error rates and small number of FLOPs. When the two algorithms are compared in terms of font cardinality, one can see that the MOSA is able generate more solutions. When the MOSA fronts are compared to the RS front, it is observed that each of three MOSA front yields in better spread value than the RS front. But for other metrics, while the best value is always achieved by a MOSA front, the RS yields in competitive results. As this front analysis is important in terms of providing some indications about the search ability of the algorithms, a more reliable comparison can be made after training the solutions in the trade-off front for longer epochs in order to get their accuracy on the original test set. However, due to the computational cost of this training process, only the selected solutions are allowed to run for longer training epochs. When the MOSA and the SA configurations are compared after this long training process,

the results show that the SA performs slightly better than the MOSA under single-objective setting. However, when the complexity in terms of both FLOPs count and the number of parameters is considered, the MOSA solutions are superior to SA solutions. When it comes to comparison to the RS solutions, the results suggest that all RS solutions are dominated by at least one MOSA solution. The MOSA configurations are also compared to the configurations obtained by other search methods like evolutionary algorithms and reinforcement learning methods. Although the results might suggest a poor MOSA performance in terms of test accuracy, the other objective, FLOPs, should also be taken into account for a fair comparison. Moreover, most of these approaches include complex augmentation strategies in order to boost the final test accuracy. When both the test accuracy and the complexity are considered, it is shown that the MOSA configurations are not dominated by any of those architectures, and the proposed method can be of great use when the computational complexity is as important as the test accuracy. It can also be concluded the MOSA which is a single-stage algorithm is able to generate high quality solutions under limited computational resources.

## CONCLUSIONS

In this study, we model a CNN hyper-parameter optimization problem as bi-objective optimization problem considering two competing objectives, namely, the classification accuracy and the computational complexity which is measured in terms of the number of floating point operations. For this bi-criteria hyper-parameter optimization problem, we develop a MOSA algorithm with the aim of obtaining high-quality configurations in terms of both objectives. CIFAR-10 is selected as the benchmark dataset, and the MOSA trade-off fronts obtained for this dataset are compared to the fronts generated by a single-objective SA algorithm with respect to three front evaluation metrics. The results show that the MOSA algorithm is able to search the objective space more effectively than the SA method. Some non-dominated solutions generated by both the MOSA and the SA search processes are selected for longer training in order to obtain their actual accuracy values on the original test set. The results again suggest that the MOSA performs better than SA under bi-objective setting. The MOSA configurations also compared to the configurations obtained by other search methods like population-based algorithms and reinforcement learning methods. It can be concluded that the MOSA which is a single stage algorithm is able to generate high quality solutions under limited computational resources, and it can be of great use when the computational complexity and time are as important as the accuracy.

### Funding
The authors received no funding for this work.

### Competing Interests
The authors declare that they have no competing interests.

## Author Contributions

- Ayla Gülcü conceived and designed the experiments, performed the experiments, analyzed the data, performed the computation work, prepared figures and/or tables, authored or reviewed drafts of the paper, and approved the final draft.
- Zeki Kuş conceived and designed the experiments, performed the experiments, analyzed the data, performed the computation work, prepared figures and/or tables, authored or reviewed drafts of the paper, and approved the final draft.

## Data Availability

Code and raw results are available at GitHub: https://github.com/zekikus/MOSA-cnn-hyperparams-optimization.

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
