# Peer review of "Multi-objective simulated annealing for hyper-parameter optimization in convolutional neural networks"

_PeerJ Computer Science, doi:10.7717/peerj-cs.338_

## Round 0.1 · original submission · Major Revisions

Both reviewers believe that the paper has some merits, but also that some additional work is necessary to make it significant enough (w.r.t. previous work of the same authors). Therefore, the paper in the present form cannot be accepted, but the authors are strongly encouraged to improve it, along the lines traced out by the reviewers (in particular, rev. 2) and resubmit it.

Reviewer 1 ·

Basic reporting

The language is mostly clear and easy to understand. Throughout the paper, there are some minor mistakes regarding grammar, phrasing or word choice (e.g. a missing article in line 2), which could benefit from having a native speaker proof-read the text.

Relevant literature is cited and sufficient context is provided. Some more references in lines 28-35 would be helpful (or, alternatively, relating matching references in the following sentences to the examples given in these lines).

Figure 3 (SA-pseudocode):
Since the acceptance criterion is an important and mostly unique feature of SA, I would recommend including it in the pseudocode directly.

Figure 4:
The three subfigures use different axis scales, which makes comparison difficult.
It is also not helpful that the symbols for the SA final front are different between subfigures 3b and 3c.
I would prefer more detailed figure captions that already describe the difference between initial and final fronts without having to refer to the text.

Experimental design

The topic of the article undoubtedly fits within the scope of the journal and the research question is clearly stated and relevant.

Section 3.1:
The allocation between inner and outer loop repetitions is usually irrelevant for SA, as long as the cooling factor is adjusted accordingly - however, if this is intended to be shown experimentally, two arbitrarily chosen configurations seem insufficient to me.
More important is the choice of cooling factor or minimum temperature (choosing one determines the other given a fixed budget of moves), which is not mentioned at all here.

An important part that is missing is a description of or reference to the single-objective SA algorithm used (some parts can be inferred from Section 4, but a definition should certainly be given alongside the algorithm description of MOSA).

Validity of the findings

The authors use publically available benchmark datasets for their evaluation, which are also referenced in the article.
The source code is available online, which is very positive!
It would be great if the raw evaluation results could also be posted in that repository.

Section 3.3:
I don't think that reporting average objective values is useful here - the whole point of Pareto-fronts is that the quality of a set of solutions cannot be easily expressed as a single value.
Section 3.3.4:
It seems like the state-of-the-art models used have the same structure across all three benchmark datasets, while the model produced by MOSA was different for each dataset - if this is the case, it should be mentioned, as this would give an advantage to the MOSA-model.

Given that the main limitation for more definitive conclusions is the low number of solutions on the final fronts, evaluations with more iterations would be quite interesting.

I also think that the article would benefit from a comparison with other state-of-the-art algorithms for automated CNN architecture generation (as referenced in Section 1) in order to be able to judge the performance of MOSA and SA compared to population-based approaches.

Additional comments

A short description of the terms Pareto-optimal and Pareto-front would be beneficial (in particular since multi-objective optimization is described - anyone unfamiliar with MOO is unlikely to know these terms).

Section 2:
In the paragraph starting at line 120, several formulas or variables seem to be missing in the text.

The role and functionality of the different layers is not easily understandable (maybe also due to missing formulas); section 2.1 could be expanded a bit. In particular, the types of layers (convolution, pooling, strive, fully connected,...) mentioned differ between the text, the attached figure and the table listing the parameters (Table 2). The same happens in Section 2.3, where yet another set of different layers or components is used.

Reviewer 2 ·

Basic reporting

The work is well structured, and mostly written in a fluent and correct English (bar few typos). The code is provided as GitHub repo, though it is not mentioned in the paper.

Some typos:
- line 39: NAS, which -> remove which
- lines 81-82, does it refer to Lu et al. or Gulcu and Kus?
- line 114: There might be multi fully connected... -> multiple
- line 208: neighbouhood -> neighbourhood

Experimental design

Motivation for the choice of the algorithm aside, the description of the MOSA algorithm is very generic, and impossible to replicate. It is not explained how a new solution is generated, nor any parameter is given (initial and final temperature value, cooling rate, temperature length, etc.). The same applies for the single-objective SA (whose objective function is stated only in Section 4).

The experimental section lacks details and clarity too. How many tuning experiments have been performed? There is no statistical analysis of the results. Which is the computational environment, and which libraries and software packages have been used? How long does it take to perform a full run of MOSA? NAS is a very trendy topic at the moment (see the list at [1]), and the rush to publish papers on this topic may lead to inconsistent experimental practices. Please try to follow the checklist described in [2], which is mostly valid also when there is no comparison against other NAS methods.

Some descriptive parts are a bit unclear, such as the introduction to multi-objective optimization (Section 1, lines 54-63), the description of CNNs (Section 2.1, where the inline math notation is also missing), or the solution representation (Section 2.3). A bit more explanation would be helpful in particular for readers not fully acquainted with CNNs.

There is no explanation about the choice of hyperparameters or their ranges. The authors restrict the number of hyperparameters and their value by "taking into account two criteria, namely computational complexity and classification performance" (Section 2.2, lines 132-134). How is this restriction done? Are all the parameters categorical? For example, in Table 2 there is no mention of the number of layers, how many are used? The authors refer to their previous work (Gulcu and Kus 2020) but a short explanation should also be included in the present paper.

References

[1] https://www.automl.org/automl/literature-on-neural-architecture-search/

[2] Best Practices for Scientific Research on Neural Architecture Search, Marius Lindauer and Frank Hutter, https://arxiv.org/abs/1909.02453

Validity of the findings

This work is very similar to a recent paper by the same authors (cited as Gulcu and Kus 2020), where a variant of SA was used to automatically obtain a CNN. The main difference is that the present work models the NAS task as a bi-objective optimization problem, where the two objectives are classification accuracy and complexity of the network, in terms of the number of parameters obtained. It is however not very clear to me what is the precise goal of this paper, whether to demonstrate the feasibility of the task (some previous works already model NAS as bi-objective optimization problem), or to propose a new algorithm for NAS (there are no sufficient details and experiments neither to claim a new state of the art, nor to properly assess the method). Two additional questions: (i) it seems to me that the MOSA algorithm is generic enough to tackle any NN task, with only the set of N hyperparameters and the data to discriminate, why focusing on CNNs? and (ii) why a multi-objective variant of Simulated Annealing, and not a different algorithm?

To assess the results of the NAS method, it would be interesting to also observe how it performs against LeNet, which is used by the authors as starting solution of their MOSA. The discussion is also short and not very deep. In particular, there is no assessment of the impact of MOSA on the final networks obtained, nor any description of the final networks. On the contrary, the authors state (Section 4, lines 373-374) that the short trainings employed by MOSA can be misleading. In other words, it is not clear whether, or up to which extent, the results reported are due to MOSA or to the longer training of LeNet.

There is no comparison against other NAS methods.

Additional comments

In this work, the authors propose a multi-objective Simulated Annealing (MOSA) algorithm to generate Convolutional Neural Networks (CNNs) taking into account two objectives, accuracy and the number of parameters of the network.

This work belongs to a thriving field of research, namely Neural Architecture Search (NAS, the AutoML task aimed at automatically designing and configuring neural networks), and is perfectly within the journal scope.

I have, however, some concerns about this work, about some unclear descriptions (including the objective of this work), a lack of details about implementation, and a weak experimental section and analysis. The list of concerns is reported in the sections above.

Overall, I do not feel this work can be published in the current state. It is however difficult to assess the performance of the proposed method, as the experiments reported do neither support nor reject its validity.

Other minor comments:

A short description of the three benchmark should also be included.

There is at time a possibly confusion when using the term "parameters", which refers to different things, depending on the level we are operating (e.g. in line 26). Since in ML the parameters are usually the weights, I suggest to maintain a consistent division between "parameters" (the weights of the NN), "hyperparameters" (those optimized by the MOSA), and "SA parameters" (temperature, etc).

Some citations are missing for the works mentioned in the literature review (RS, BO, etc, lines 28-36).

---

## Round 0.2 · Minor Revisions

Both reviewers believe that the paper has improved significantly. However, they also believe that the presentation needs some extra cleaning and small improvements in order to be acceptable. I agree with them and I would ask the authors to do a new round of revision along the tracks well marked by the reviewers.

Reviewer 1 ·

Basic reporting

The language is improved in the revision. Some mistakes remain, mostly missing articles and prepositions (e.g. in line 13: "we pose 'a' CNN hyper-parameter [...] as 'a' bi-criteria [...]"

Missing literature references have been provided. Since this work seems to build at least partially on prior work by the same authors, the previous paper should already be mentioned in the introduction.

The structure of the article could use some further revision to ensure that each section contains all relevant information. In particular, the descriptions of the algorithms used are distributed across multiple sections (Section 3.2.3 should be part of the algorithm description, the description of SA used in this paper appears piecewise in 3.3.1 (552ff) and 4 (624f)). Section 3.2.4 should already state which solutions are selected for longer training (given later in 3.3.2).

Figure 6: The difference between SA and MOSA is barely recognizable - I recommend that you use a log scale for the x-axis instead.

Experimental design

The description of the methods and configuration has been much improved in the revision.
What's still missing is a short description of the RS algorithm (line 540, presumably similar to SA, except that all solutions are accepted?).

The description of the neighborhood and moves could be improved. It is currently not clear, which moves exist (e.g. is "modifiying each convolution block" a single move that affects every conv. block at the same time? Or is one block chosen at random? Are the three subitems all performed sequentially or is one of them chosen? What are the weights for this choice?)
It also seems like the neighborhood parameters (the composition and weights of the different moves) have been chosen arbitrarily. Some motivation should be given for these choices (why increase the probability of adding blocks over time? Why at this rate? Why is there no move that removes blocks?)
Alternatively, including these parameters in any kind of tuning could provide better performance and robustness.

Regarding tuning:
- The initial temperature selection strategy is intended to be a reasonable guess when suitable values are not available prior to the SA run (online tuning). In this case, where offline tuning is performed, potentially much better values could be achieved, either by trying out different values manually or using an automated parameter tuner. The automated initial temperature selection strategy could be used to provide an initial starting value for this tuning.
- The ratio between inner and out loops (e.g. the cooling rate) is mostly irrelevant in this case (geometric cooling, fixed initial and end temperature, fixed iteration budget) for SA in general.

Validity of the findings

The term "competitive results" is misleading - there is a clear tradeoff between the complexity of the model (FLOPs) and the accuracy. To show actual competitiveness with the state-of-the-art, MOSA would have to generate at least one solution along the pareto front that achieves similar accuracy (which is the objective optimized in those other papers).
This is not to say that MOSA has no merit - it simply focuses on another objective criterion; it performs well for a different problem.

At the same time, there is lost potential of MOSA due to missing tuning of neighborhood parameters, and the reduction of the training dataset.
If the full CIFAR-10 dataset is used at least for the longer training runs in the final evaluations (Section 3.3.2), the impact of the latter is only minor, but the model might still give better results under the full dataset.

Another point is that you should not use the best results for MOSA and compare to median results for SA and RS. Either use median or best results for both (median is already skewed anyway, since you pick only good solutions for longer training).

Additional comments

Some minor comments regarding specific lines:

134: FLOPs is Floating Point Operations (not Floating Point Operations per Second (FLOPS)). FLOPS is also mistakenly used in line 544.

280: k depends on the number of objectives in general (not just 2)

333ff: It is not clear whether the return-to-base mechanism is applied to a random solution in the archive or only a solution that dominates X'

555: Why is SA longer than MOSA? The overall algorithm structure is the same and for comparability, training and evaluation of solutions should be performed the same way for both.

562ff: Generational Distance is already described in Section 2.4, repeating the description is not necessary here

Reviewer 2 ·

Basic reporting

The authors rewrote a large part of the paper, including several additional sections. Overall the paper reads well, but I feel one more pass to polish the text would make it more readable ( e.g. by breaking down the longer periods and adding some punctuation).

At times there is a wrong spacing between in-line formulas and the following text (with space missing) or punctuation (with space, to be removed), e.g. lines 103-104.

Experimental design

The authors now include more details about the algorithmic choices and the experimental setup. While some of these choices are motivated (such as the initial and final temperature values, analyzed thoroughly), other ones seem arbitrary, such as the probabilities for selecting an operator in the local moves. The same applies for the analysis of the cooling rate: only three values are evaluated with three experiments each. The authors report that a statistical test showed no differences, but with only 3 experiments for each value it's hard to expect anything different. More experiments (e.g. 15 for each value, or another value determined from the beginning) might show a different outcome. Another potential issue with the experiment reported is that, for a short MOSA run, a cooling scheme of 0.85 might be still too high and allow too much diversification in the beginning, therefore spending too much time on poor quality solutions, while a lower cooling rate (e.g. 0.4, or 0.6) could immediately focus towards better quality solutions.

The hyperparameters of SGD are often included in this kind of studies, why are they left out here?

The underlying issue here is the extremely expensive function evaluation of each MOSA iteration, for which there is no real shortcut. The authors report 50 hours for one MOSA run on their machines, so it is understandable that they try to minimize the overall time. Unfortunately, too few experiments will be uninformative.

Validity of the findings

The same applies for the comparison with SA and RS reported in Figures 4 and 5: the frontiers are very close, there is probably no statistical difference. But again, this may be a limitation due to the few experiments considered.

The authors now report a comparison with other methods (we can cavil about implementation details, but that's essentially the goal of NAS: automatically configure a neural network), which are reported to have much lower error rates, at the cost of a much higher complexity of the resulting NNs. It would be interesting to see a discussion about the resulting networks from the various methods: the network number 1 obtained with MOSA has a comparable number of parameters than other NAS methods who have higher accuracy, but obtained at a much higher computational cost. Does this mean the networks found with MOSA are sparser?


Overall, it appears that MOSA improves over its starting solution, and obtains results that can be compared in some terms with other methods; it is less clear whether it really improves over the SA previously proposed by the same authors.

Additional comments

I thank the authors for addressing my comments on the first version of this paper. I think the comparisons now reported give an indication about the potential of the method, but more experiments are needed to fully appreciate the extent of the improvement over existing methods (including the SA of the same authors).

Minor:
- line 46: it is shown in many studies... : there should be some citations here
- line 172: an input is not necessarily an image. Aside form other matrix/tensor inputs, 1-D CNNs can be used for example for time series.

---

## Round 0.3 · Minor Revisions

The reviewers believe that the paper is ready to be accepted, but that the English writing still needs some cleaning. Please, do your best to perform these grammar improvements and resubmit. I will check it quickly myself without passing it to the reviewers again.

Reviewer 1 ·

Basic reporting

The authors have adequately addressed my concerns regarding the structure.
The english has been improved in several places. In my personal opinion, the writing could still benefit from being read by a native speaker, but it is certainly clear enough to follow.

The figures are much clearer now, log-scale was certainly the right choice for all three of them.

As a minor point on formatting, I would recommend to add some spacing and a uniform style for the move descriptions in 3.1.2

Experimental design

The description of the moves is clear after the revision.

Many of the values and the chosen operations themselves still seem arbitrarily selected, but I recognize that in-depth tuning is practically infeasible.
Unfortunately, the low number of evaluations and the low number of iterations makes it hard to determine how and by how much these parameters influence the quality of results.
I suspect that SA in general may not be the best approach for this kind of problem, as it typically derives its strength from fast applications of large numbers of moves (millions, if not billions). The same algorithm using the Metropolis criterion at a fixed (low) temperature or even random walk accepting only improving solutions might result in the same or even better solutions, as reaching a local optimum in the given number of iterations is unlikely anyway and these simpler alternatives would be easier to tune and waste less time on worsening moves.
The same holds for the structure of the neighborhood, which is quite complex and large for SA. Breaking this up into multiple, simpler types of moves would allow a more in-depth analysis of the impact of each type of change and more focussed tuning - however, it would require proportionally more iterations to reach the same solutions, which of course is undesirable here due to the large evaluation cost.
Of course this does not invalidate the experimental setup of the paper, but it should be kept in mind when attributing the success of the algorithm to any specific component (except the multi-objective setting, which is independent of the solution method itself).

Validity of the findings

As mentioned before, the statistical significance of the results is hard to determine with the given number of evaluations, in particular when comparing between MOSA, RS, and SA.
Certainly more experiments in the future would help in determining the best approach for this problem (see also my comments above).

Additional comments

No further concerns.

Reviewer 2 ·

Basic reporting

There are still some imprecision in the grammar, such as "We pose A CNN hyper-parameter optimization problem..." in the conclusion, and some missing articles in front of the method names (MOSA and SA) throughout the whole paper.

Experimental design

No comment

Validity of the findings

To overcome the main limitation of NAS studies, namely the computational complexity of the task, I was recently made aware of this work: https://arxiv.org/pdf/2008.09777.pdf

The authors provide a surrogate benchmark of neural architectures on CIFAR-10, so that the evaluation of an architecture becomes way cheaper.

This is a preprint that was released few months ago, after the first submission of this paper, so of course it is not to be taken into consideration for this paper. However, should the authors pursue this line of research, I recommend they look into this benchmark, to be able to perform statistical evaluations of their (and other authors') methods.

Additional comments

Aside from some minor correction in the writing, I have no further comments on this paper.

---

## Round 0.4 · accepted · Accept

I am happy with the latest revision.